# Combination of Nanobioproduct and Chemical Ethylene Synthesis Inhibitor with Entomopathogenic Fungi: A Novel Management Strategy for Coffee Berry Borer in Arabica Coffee

**DOI:** 10.3390/plants14101495

**Published:** 2025-05-16

**Authors:** Lilian F. Sousa, Ana P. A. Antunes, Maísa M. Moreira, Érika H. Arantes, Ezequiel G. Souza, Bruno H. S. Souza, Tatiana Cardoso e Bufalo, Camila G. Freitas, Caroline Dambroz, Joyce Dória

**Affiliations:** 1Department of Entomology, Federal University of Lavras (UFLA), Lavras 37203-202, Brazil; fslilian@outlook.com (L.F.S.); ana.antunes5@estudante.ufla.br (A.P.A.A.); maisa.moreira2@estudante.ufla.br (M.M.M.); erika.arantes@ufla.br (É.H.A.); ezequiel.souza@estudante.ufla.br (E.G.S.); brunosouza@ufla.br (B.H.S.S.); 2Department of Physics, Federal University of Lavras (UFLA), Lavras 37203-202, Brazil; tatiana.cardoso@ufla.br; 3Department of Agriculture, Federal University of Lavras (UFLA), Lavras 37203-202, Brazil; camila.garciafreitas@gmail.com (C.G.F.); carolinedambroz@gmail.com (C.D.)

**Keywords:** coffea arabica, *Hypothenemus hampei*, integrated pest management, *Bacillus subtilis*, *Beauveria bassiana*, *Metarhizium anisopliae*

## Abstract

Brazil is the leading producer and exporter of coffee, accounting for more than one-third of global production. However, the coffee berry borer (CBB), *Hypothenemus hampei*, poses a significant threat to coffee yield and quality. Its control has been primarily based on the use of chemical insecticides, which entail risks to human health and the environment. We evaluated the efficacy of an ethylene synthesis inhibitor and a *Bacillus subtilis*-based nanobioproduct combined with entomopathogenic fungi against the CBB. The treatments included combinations of potassium phosphate-based and nanobioproduct-based bioregulators, bioinsecticide comprising *Beauveria bassiana* + *Metarhizium anisopliae*, and chemical insecticides (acetamiprid + bifenthrin). The experiment included an in vitro assay assessing the reproduction of CBB females on treated coffee berries and a field trial evaluating the impact of the treatments on coffee bean quality, beverage sensory attributes, and antioxidant enzyme activities. All insecticide treatments, except the bioregulator at 6 L ha^−1^ + insecticides, significantly reduced larvae production. The 6 L ha^−1^ bioregulator treatment reduced bean defects. The fungi + insecticide treatment increased superoxide dismutase activity, while ascorbate peroxidase activity was highest in the control, followed by the fungi + nanobioproduct treatment group. The nanobioproduct bioregulator combined with entomopathogenic fungi proved to be an efficient strategy for managing the CBB.

## 1. Introduction

Coffee ranks as the world’s second most valuable commodity, considering the entire production chain, from harvest to the coffee cup of the end consumer [1]. Brazil leads as the largest coffee producer and exporter, contributing more than one-third to global production, with 54.2 million bags (60 kg) produced [2]. Despite this global leadership, Brazil faces significant yield and quality losses due to insect pest attacks, particularly by the coffee berry borer (CBB), *Hypothenemus hampei* (Coleoptera: Curculionidae: Scolytinae), which is considered the primary pest of coffee crops worldwide [1,3]. The CBB is responsible for estimated annual losses of around USD 215 to 358 million [4,5].

CBB is a monophagous insect native to Africa that feeds and reproduces on coffee beans, boring into them to lay eggs, after which the ecloded larvae cause injury as they develop inside the beans. This not only reduces the quantity of harvested coffee but can lead to the depreciation of beans, reducing their quality for commercialization [5,6]. Controlling this pest represents a significant challenge, and chemical insecticides are still widely employed as the primary control method. However, it is important to consider the negative impacts that this approach can pose to both the environment and human health [6].

To find sustainable alternatives for the chemical control of CBB, the implementation of integrated pest management strategies, including cultural, biological, and behavioral methods, has received increasing attention. Potential pest management strategies have explored the use of bioregulator products that act as ethylene synthesis inhibitors [5]. Ethylene (ET) is one of the main phytohormones involved in signaling pathways that mediate plant resistance to insects, along with other phytohormones, especially jasmonic acid (JA) and salicylic acid (SA) [7]. ET interacts with jasmonic acid, salicylic acid, and abscisic acid (ABA), often acting synergistically with JA and directly influencing insect feeding and development. For instance, the suppression of ET signaling has been shown to enhance SA biosynthesis, playing a pivotal role in the activation of distinct defense pathways that are particularly effective against piercing–sucking insects. There are other instances where JA signaling also trades off with that mediated by ET, interfering with responses against chewing insects and necrotrophic pathogens, respectively. According to Pieterse et al. (2012) [8], ET affects and finetunes the outcome of the JA response, which in turn is more specific against chewing herbivorous insects like the coffee berry borer. For instance, ET can act synergistically on the ERF branch of the JA signaling pathway, but can also antagonize the MYC branch of the same JA pathway that usually participates synergistically with ABA. Therefore, modulation of the JA-mediated induced defense responses by ET signaling can negatively affect host plant resistance to insect attack [9]. In this sense, suppression of ET synthesis could increase JA signaling and defense genes expressed by JA against the coffee berry borer. The application of ethylene synthesis inhibitors during the CBB flight period can reduce the attractiveness of fruits to colonizing CBB females since they prefer more mature fruits for oviposition due to the higher dry matter content. Furthermore, by prolonging the maturation period of coffee fruits, the bioregulator can reduce the pest infestation window, exposing the females to other control methods and environmental adversities [5,10].

Microorganisms are increasingly used in agriculture, not only for inducing plant defenses but also for their use in the biocontrol of pest insects and phytopathogens [11]. For instance, the fungus *Trichoderma* spp. has been shown to be effective against pathogens and insects, although the registration of commercial products is only focused on plant disease control [12]. In the control of the CBB, *Beauveria bassiana* is the major microorganism used in Brazil and worldwide, but its efficiency is affected by climatic conditions, requiring high humidity and low solar radiation to optimize its action [1]. The bacterium *Bacillus subtilis* is gaining prominence in promoting both plant growth and direct and indirect disease biocontrol, causing both antimicrobial effects and plant resistance induction. However, these microorganisms are vulnerable to solar radiation, necessitating the use of photoprotective products and coatings to maintain their efficacy, especially in agricultural practices [13]. Thus, we highlight the importance of studying microorganism coating with nanoparticles as a means of solar protection and increased biocontrol.

In this context, nanotechnology emerges as a promising approach [14]. By applying an innovative development process such as the green synthesis of nanoparticles (NPs), more sustainable pest control methods are promoted, capable of mitigating the adverse environmental impacts associated with conventional pesticides, while potentially enhancing crop productivity [15]. Coating microorganisms with specific NPs enables them to survive, even when applied during hot periods of the day, performing functions that range from increasing biotic and abiotic stress resistance to nutrient availability [16].

Considering the importance of coffee cultivation in Brazil and other countries of the world, the necessity of making agriculture more sustainable in light of climate change, and the need to manage pests without impairing coffee productivity and beverage quality, the present work aimed to evaluate the performance of bioregulators acting as ethylene synthesis inhibitors, such as the developed *B. subtilis*-based nanobioproduct and a commercial chemical bioregulator combined with entomopathogenic fungi, compared to insecticides on CBB reproduction and injury under in vitro conditions, as well as performance in the field regarding coffee bean physical characteristics, beverage quality, and antioxidant enzyme activities.

## 2. Results

### 2.1. Biological Development Bioassay with Hypothenemus Hampei

In the present study, different treatments were evaluated, consisting of combinations of a chemical ethylene synthesis inhibitor, chemical and fungi-based insecticides, and a developed ethylene synthesis inhibitor nanobioproduct, on CBB reproduction and injury to coffee beans.

No significant difference (X^2^ = 1.95; *p* < 0.8565) was observed for the number of eggs of the CBB in coffee fruits treated in the laboratory, indicating that the applied treatments did not impact the oviposition of females after contact with the treatments (Figure 1).

A significant difference (X^2^ = 64.02; *p* < 0.0001) was observed among treatments in the numbers of CBB larvae produced after 30 days in arabica coffee fruits treated with immersion in the laboratory. The treatments with the application of insecticides (T5), bioregulator 6 L ha^−1^ + insecticides (T7), and insecticides + fungi (T8) significantly affected the number of CBB larvae produced by females exposed to the treated fruits. On the other hand, the bioregulator 3 L ha^−1^ + fungi treatment (T3) resulted in a higher number of larvae produced compared, thus favoring CBB reproduction (Figure 2).

Based on the calculation of larval reduction efficiency (%), treatments combined with insecticides showed high rates of CBB larvae reductions, achieving 100% reduction efficiency in those treatments, except for bioregulator 3 L ha^−1^ + insecticides, which had 64.44%. Additionally, the combination of fungi + nanobioproduct showed promising results regarding the reduction in CBB larval development, with 97.68% efficiency (Figure 3).

### 2.2. Field Experiment

#### 2.2.1. Physical and Sensory Analysis of Coffee Beans

No significant differences were observed for the parameter score (X^2^ = 13.54; *p* = 0.1395), acidity (X^2^ = 2.90; *p* = 0.9680), body (X^2^ = 3.33; *p* = 0.9499), sweetness (X^2^ = 5.16; *p* = 0.8204), finish (X^2^ = 7.49; *p* = 0.5862), and type (X^2^ = 0.87; *p* = 0.997) of coffee beans. On the other hand, a significant difference (X^2^ = 17.54; *p* = 0.0409) was observed for the coffee bean defects in treatment 2 compared to the control (Table 1).

Based on the reduction efficiency (%) of coffee bean defects, all products demonstrated high rates of reduction compared to the control. Additionally, the treatment with bioregulator 6 L ha^−1^ showed a superior reduction in bean defects compared to the other treatments, with a 43.24% reduction relative to the control (Figure 4).

#### 2.2.2. Enzymatic Analysis

A significant difference (*p* < 0.0001) was observed for the enzymatic activity of SOD among treatments. The treatments with insecticides + fungi (T8) and fungi + nanobioproduct (T9) showed higher enzymatic activity compared to the other treatments, with SOD activity being approximately 1959%, and 669% higher, respectively, than the average of the other treatments (Figure 5).

A significant difference (*p* < 0.0001) was observed for the enzymatic activity of APX. The control treatment (EC) showed the highest enzymatic activity, followed by nanobioproduct + fungi (T9), differing significantly from the other treatments. The activities of APX in these treatments were 1873% and 996% higher, respectively, than the average of other treatments (Figure 6).

## 3. Discussion

In the present study, the application of chemical and biological ethylene synthesis inhibitors mixed with entomopathogenic fungi compared to chemical insecticides was evaluated as a potential management strategy of the CBB in arabica coffee. The direct effects of the treatments on CBB reproduction in laboratory-treated fruits were recorded, as well as the indirect effects on the physical and sensory characteristics, and activities of antioxidant enzymes in the coffee beans collected from the treated-plants in the field. It is worth noting that as a biological ethylene synthesis inhibitor, we used *B. subtilis* bacteria coated with NPs through a green synthesis methodology previously developed by our research group [17].

Under controlled laboratory conditions, treatments involving combinations of chemical insecticides did not exert a direct effect on the oviposition behavior of CBB females. In such instances, the observed reduction in egg deposition—and consequently in larval eclosion—was most likely attributable to the mortality of adult females following exposure to the treated berries. It is important to highlight that the action of the insecticide acetamiprid, belonging to the neonicotinoids group, works by binding to the nicotinic acetylcholine receptors in the peripheral and central nervous systems. This process alters the membrane potential, interrupting nerve transmission and causing excessive excitation. This disruption ultimately leads to paralysis and, eventually, the death of the insect [18]. On the other hand, bifenthrin, an insecticide from the pyrethroids group, exerts its action primarily on sodium channels, which control the passage of ions across the axon’s cell membrane. It lowers the threshold for action potential generation in nerve and muscle cells, resulting in repeated stimulation. At high concentrations, the increased sodium influx can block action potential generation, disrupt nerve conduction, and cause paralysis [19].

The treatment using the chemical bioregulator at a dose of 3 L ha^−1^ combined with fungi caused an antagonistic effect, resulting in higher numbers of CBB larvae. The same treatment also presented numerically higher egg numbers. This antagonistic response was unexpected, and we do not know the exact mechanistic process behind this outcome. A potential hypothesis that could explain this result may involve hormetic-like responses. Hormesis is a widespread phenomenon in several living organisms characterized by biphasic dose–response curves in which low doses of a given chemical or stress induce stimulatory effects, whereas high doses promote inhibition [20,21]. For instance, plants exposed to low levels of stress can be better protected through stimulation of cellular defense mechanisms that aid in homeostasis control [22]. The same biphasic dose–responses are reported for insects against chemical insecticides and transgenic plants, favoring resistance evolution in pest populations [23,24]. Therefore, we can hypothesize that CBB females exposed to stress provided by the chemical bioregulator at q lower dose combined with entomopathogenic fungi triggered stimulatory responses in the CBB that benefited larvae production and survival.

In the bioassay with the CBB using coffee fruits treated by immersion in the treatment solutions in the laboratory, the treatments containing the chemical ethylene synthesis inhibitor alone or combined with entomopathogenic fungi showed low efficiency in reducing CBB reproduction. This result was distinct from those obtained in a previous study where the same bioregulator (Mathury^®^) applied in arabica coffee fruits provided lower CBB infestation in the field and lower preference and performance in the laboratory [5]. These differences in results between the studies should be a function of the use of coffee fruits treated by distinct application methods, which herein involved immersion of the fruits and treating the fruits while in plants in the field. Treating the fruits directly by immersion in solutions with the chemical ethylene synthesis inhibitor may have provided an inverse effect that benefited the CBB.

The treatments involving combinations with chemical insecticides exhibited great efficiency in larvae reduction, which, as in the evaluation of the number of eggs, occurred due to the mortality of CBB females exposed to the treated fruits, who did not come to oviposit in the beans. Likewise, the ethylene synthesis inhibitor nanobioproduct combined with entomopathogenic fungi also provided an almost 100% larvae reduction. Although the results obtained indicate negative effects on pest development, it is important to emphasize that, due to the destructive nature of the assessments and the insect’s cryptic behavior, it was not possible to continuously and non-invasively monitor the biological events responsible for these effects. We hypothesize that the compounds applied, particularly the nanoparticles present in the nanobioproduct, may have directly affected the reproductive system of adult insects, potentially interfering with oogenesis and thereby compromising their reproductive capacity. Alternatively, the effects may be related to egg inviability and disruption of embryonic development during the early stages. Moreover, regarding potential direct effects on larvae and adults developing inside the fruit, it is well established that conventional chemical formulations often face limitations in reaching such targets due to physical barriers—both the entry point of the fruit and the insect’s own protective structures, such as the larval integument. In contrast, given the nanometric scale of the particles in the nanobioproduct, it is plausible to hypothesize that these structures, owing to their reduced size and greater mobility, may possess enhanced penetrative and diffusive capabilities, enabling more effective and direct action against internal developmental stages of the insect.

It is noteworthy that the effects on CBB infestation in the fruits in the field were mainly evaluated via the plant, while in the laboratory, the application was made by immersion of fruits in solutions, thus evaluating direct effects on the pest. In this way, the high efficiency of CBB larvae reduction in fruits treated with fungi + nanobioproduct is very interesting in the context of integrated pest management. Although *B. subtilis* bacteria also inhibit ethylene synthesis, its physiological mechanism differs from that provided by the chemical bioregulator (Mathury^®^) used in this study. *B. subtilis* is associated with the induction of resistance in plant tissues through the production of ACC deaminase (ACCd). ACCd suppresses ethylene production by sequestering one of its main precursors, 1-aminocyclopropane-1-carboxylic acid (ACC). The presence of ACCd triggers higher H_2_O_2_ production, thus stimulating the induction of defense compounds, including the phytohormones jasmonic acid and salicylic acid. Moreover, increased H_2_O_2_ concentrations can cause damage to insect cells, and ethylene suppression can induce the production of secondary compounds and defense enzymes, such as protease inhibitors and kinase blockers, which directly affect insect development [25,26].

Concerning the effects on the quality of the coffee produced, the beans from plants treated in the field were evaluated for their physical and sensory characteristics after harvest. It was found that the application of the treatments did not influence the score, acidity, body, sweetness, finish, and type parameters. However, defect classification, including defects caused by CBB infestation in the beans, was significantly influenced, so that the isolated chemical inhibitor at 6 L ha^−1^ reduced defects by ~43% relative to the control. The chemical ethylene synthesis inhibitor (Mathury^®^) consists of potassium acetate, which acts by inhibiting ethylene synthesis, prolonging the photosynthetic activity of the leaves. It reduces leaf senescence, allowing the leaves to remain on the plant for longer and producing photoassimilates. This mechanism is essential for fruit and grain filling, as well as for improving post-harvest quality [27].

Similar results were obtained by Martins et al., 2023 [5], who evaluated the same chemical ethylene synthesis inhibitor applied to arabica coffee in the field and laboratory. The authors observed that the application of a higher dose (15 L ha^−1^) of the bioregulator in the field reduced infestation in fruits, making them less preferred by the CBB. In our study, the ethylene synthesis inhibitor nanobioproduct presented a similar percentage (16.22%) of defect reduction to treatments containing insecticides.

The activities of the antioxidant enzymes SOD and APX were found to correlate to with negative effects on the CBB, as these enzymes are considered chemical markers of plant resistance to insects [28]. Higher SOD activity occurred with the application of fungi + insecticides, while APX activity was higher in the control treatment, followed by fungi + nanobioproduct. Antioxidant enzymes contribute to greater scavenging of reactive oxygen species (ROS), which play a key role in restoring and maintaining cellular homeostasis [29]. It is possible to improve plants’ antioxidant activities through the use of elicitor compounds or bioactivators, which can even act as a pest management strategy [30,31]. The results of greater APX activity with fungi + nanobioproduct can be explained by the greater accumulation of H_2_O_2_, due to the action of ACCd produced by *B. subtilis* bacteria, and consequently greater activity of the antioxidant system to remove the ROS formed [25,26].

Although the analyses of antioxidant enzymes were performed on coffee beans collected from field-treated plants, it is hypothesized that the negative effects on CBB larval development in the laboratory may have been due to the toxic action of ROS. This issue deserves further investigation into the mechanisms of action. In the case of higher SOD activity with fungi + insecticides, it was not possible to reach a logical hypothesis about such an effect, but in some way, this mixture of treatments may have provided a higher level of stress in the fruits and coffee beans, increasing ROS concentrations and consequently inducing higher activity of the antioxidant enzyme.

According to the effects observed in this study with chemical and biological ethylene synthesis inhibitors, especially regarding the reduction in the number of larvae in laboratory-treated fruits and the physical and sensory analysis of coffee beans from field-treated plants, they indicate promise for use in strategies for CBB management. There are products available on the market based on the bacterium *B. subtilis* with registration for the biocontrol of plant diseases [32,33]. The use of *B. subtilis* can allow increased crop productivity due to reduced losses from pathogen infection. Moreover, this bacterium can be found as a growth-promoting rhizobacteria and epiphytic and endophytic species [34,35,36]. However, so far, there are no registered products for the biocontrol of pest insects based on *B. subtilis*. For the CBB, currently, the main biological control agent available in Brazil and several countries is the entomopathogenic fungus *B. bassiana* [6,37,38]. In Brazil, there are thirty-nine registered products based on *B. bassiana* for CBB control, most of which are composed of the fungus alone, and two containing *M. anisopliae* in mixture. These are the only commercial products registered as microbial insecticides for CBB control. The association of microorganisms with NPs makes them less vulnerable to solar radiation, allowing them to perform more effectively [13]. Thus, the results found in the present work with fungi + nanobioproduct are promising for the development of new products for CBB biocontrol.

## 4. Materials and Methods

### 4.1. Treatments and Experimental Conditions

The treatments in this study were designed to test the hypothesis that the application of the ethylene synthesis inhibitor bionanoproduct composed of *B. subtilis* coated with zinc oxide (Zn) NPs in combination with the entomopathogenic fungi *B. bassiana* + *M. anisopliae* 1 × 10^9^ CFU mL^−1^ (DuoFunghi^®^, Satis, Araxá, Brazil) reduces CBB infestation in arabica coffee fruits in comparison to a commercial chemical ethylene synthesis-inhibiting bioregulator (Mathury^®^, Satis, Araxá, Brazil) associated with the same fungi. To assess this, product combinations were assigned to eight treatments, in addition to commercial chemical insecticides based on acetamiprid + bifenthrin (Sperto^®^, UPL, Ahmedabad, Gujarat, India) registered for CBB control (positive control) and an experimental control (negative control) using distilled water (Table 2). The chemical ethylene synthesis-inhibiting bioregulator was used in different doses, and the products were applied in some combinations to assess their possible synergy. The doses of the commercial products were used according to the manufacturers’ recommendations (Table 2).

The experiment was conducted in two stages: an in vitro test that evaluated female CBB reproduction in coffee fruits subjected to different treatments; and the application of the same treatments under field conditions to assess their influence on the physical quality of the coffee beans, the sensory quality of the beverage, and the activities of antioxidant enzymes.

In the in vitro assay, healthy coffee fruits were collected in an experimental field free of insecticide application and evaluated at the Laboratory of Plant Resistance and Integrated Pest Management of the Department of Entomology at the Federal University of Lavras, Minas Gerais, Brazil. The bioassay was conducted in a climate-controlled room under environmental conditions of 25 ± 2 °C, 60 ± 10% relative humidity, and a 12C:12D h photoperiod.

For the evaluation in field conditions, the experiment was installed in a commercial arabica coffee plantation (cv. Catuaí), located in the municipality of Lavras, MG, Brazil, between the geographic coordinates 21°18′22.6″ S and 45°03′51.7″ W. The climate of the region is classified as Mesothermal [39].

### 4.2. Preparation of Ethylene Synthesis Inhibitor Nanobioproduct

For the preparation of the ethylene synthesis inhibitor nanobioproduct, *B. subtilis* cells cultured in a minimal nutrient agar medium were transferred to 200 mL of liquid medium. The culture was scraped using a 1 µL sterile needle-tipped inoculation loop, followed by the addition of 10 mL of sterile distilled water. The resulting suspension was standardized according to the turbidity of the solution, using the McFarland nephelometric scale, and the inoculum concentration was adjusted to 1.5 × 10^8^ CFU mL^−1^. Then, 1 mL of this solution was added to 200 mL of nutrient broth in an Erlenmeyer flask.

The flasks containing the bacterial solution were maintained under agitation at 120 rpm, at a temperature of 27 ± 2 °C, with a photoperiod of 16 h over 48 h. After this period, 1 mL of the solution was pipetted into an Eppendorf tube and centrifuged at 5000 rpm for 10 min. The supernatant was discarded and the pellet at the bottom of the tube was washed twice with 1 mL of sterile distilled water. The biomass resulting from the washing was resuspended in 1 mL of sterile distilled water and added to the nanoparticle biosynthesis medium. The flasks were kept under controlled shaking and temperature.

After synthesis, the solution was filtered through Whatman No. 1 filter paper. The bacteria covered by nanoparticles retained on the filter paper were dispersed in 150 mL of distilled water. Then, the solution was diluted in 6 L of water and transferred to a manual backpack sprayer for spraying onto arabica coffee plants in the field.

### 4.3. Biological Development Bioassay with Hypothenemus Hampei

Healthy arabica coffee fruits were collected from untreated plants in the field to evaluate the possible direct effects of the treatments on CBB reproduction. The fruits were immersed in the treatment solutions (Table 2) for 10 s, allowed to dry on paper towels, and then transferred to Petri dishes (5 cm in diameter) lined with filter paper. Next, a female CBB, collected from coffee fruits in an experimental area of UFLA without insecticide application, was transferred using a fine paintbrush.

The bioassay was conducted over 30 days, and at the end, the number of eggs and larvae produced by CBB females inside the treated fruits were assessed. The experimental design was completely randomized, containing 10 treatments and 20 repetitions, with each repetition consisting of one CBB female and one coffee fruit in a Petri dish (Figure 7).

### 4.4. Field Experiment

The treatments were applied to the aerial plant parts using a manual backpack sprayer under constant pressure on mature arabica coffee plants of the Catuaí variety, at 110 and 125 days after the main flowering period. This application period was chosen as it corresponds to the flight period of the female CBB in coffee crops [40]. Additionally, a previous study demonstrated that the application of a chemical ethylene synthesis inhibitor (Mathury^®^) reduced CBB infestation in the field and negatively affected the pest attractiveness and development in the laboratory [5]. The field experiment was set up in a randomized block design, with each plot consisting of ten plants in a row, with four blocks as repetitions.

The coffee fruits were entirely harvested from plants in the plots by shaking them onto a cloth when they reached full maturity and stored in raffia bags identified by each treatment and repetition. The fruits were washed and sorted according to their ripeness: green, cherry, and raisin. The cherry fruits underwent natural drying on raised platforms in a greenhouse until they reached about 11% moisture. Subsequently, the beans were hulled and subjected to sensory analyses through cup tasting and physical classification, according to the methodology proposed by the Specialty Coffee Association (SCA) [41].

#### 4.4.1. Physical and Sensory Analysis of Coffee Beans

For the physical analysis, defects such as black, sour, green, shell, poorly formed, bored, and broken beans were counted and assigned scores ranging from 1 to 5. Scores were assigned cumulatively for the evaluated characteristics, meaning that the higher the score regarding the defect, the worse the quality of the coffee beverage.

The sensory profile of the coffee samples and the cup scores were evaluated according to the SCA, taking into account aspects of note (distinctive flavor characteristics perceived in the coffee, such as floral, fruity, nutty, or spicy notes; these notes are scored on a scale from 0 to 100, where higher scores indicate better coffee quality), acidity (bright, tangy, or crisp sensation in coffee, often associated with fruity or citrus-like qualities, reflecting the coffee’s liveliness), sweetness (pleasant sugary quality in coffee, perceived as smoothness and balance without bitterness or harshness), and finish (aftertaste or lingering flavor left in the mouth after swallowing, indicating the coffee’s complexity and clarity). The evaluations were conducted by three certified SCA Q-graders [41], using a 200 g sample.

#### 4.4.2. Enzymatic Analysis

For sample preparation, cherry coffee fruits collected from the field experiment after treatment application were washed, dried on raised platforms until they reached approximately 11% moisture, and then peeled. Subsequently, these fruits were ground in an analytical knife mill (IKA A11, Staufen, Germany), with the addition of liquid nitrogen and polyvinylpolypyrrolidone [42], and then 200 mg resulting from the grinding was weighed, placed in Eppendorf tubes, and stored in liquid nitrogen until extraction time. For the analysis, three repetitions per treatment were used for each enzyme.

The enzymatic activity of superoxide dismutase (SOD) and ascorbate peroxidase (APX) was extracted according to the methodology described by Biemelt et al. (1998) [43]. SOD activity was determined as described by Giannopolitis and Ries (1977) [44], and APX activity was determined according to the protocol of Nakano and Asada (1981) [45].

### 4.5. Statistical Analysis

The data were subjected to an exploratory analysis of the homogeneity of variance and normality of residuals, using the Levene and Kolmogorov–Smirnov tests, respectively. The data obtained in the CBB bioassay did not show a normal distribution and were analyzed using the generalized linear model (GLM) with a Poisson distribution and log-link function, and the means of the treatments were compared using Tukey’s test (α = 0.05). The data obtained from the physical and sensory analyses of the coffee beans were analyzed using general linear models with a Gamma distribution and log-link function for the score and defects and a Poisson distribution and log-link function for the other parameters, and when significant, the means of the treatments were compared using the LSD test (α = 0.05). The enzymatic data were analyzed using a GLM with a Gaussian distribution for APX and a negative binomial distribution for SOD. Subsequently, a Dunnett’s test was applied to compare each treatment with the control, considering a significance level of 5%.

For the calculation of control efficiency of treatments, Abbott’s formula (1925) [46] was used. The formula is represented as follows:E% = [(T − TR)/(T)] × 100

E: efficiency of larvae reduction corrected by the control;

TR: number of live larvae in each treatment;

T: number of live larvae in the control.

## 5. Conclusions

In this study, different treatments were evaluated, combining chemical and biological control methods and their association, in search of sustainable and efficient strategies for CBB management in arabica coffee. By assessing their effects on CBB performance and the quality of the coffee beans and beverages from laboratory and field trials, we found that the use of *B. subtilis* coated with Zn NPs combined with entomopathogenic fungi proved to be a viable and sustainable strategy for CBB management.

## 6. Patents

The methodology developed by this research group for coating *B. subtilis* bacteria with Zn NPs (nanobioproduct) resulted in the patent “System for the slow release of nanoparticles by microorganisms and/or a consortium of microorganisms and/or their virulence structures for application in biotization and/or formulations of biopesticides, biofertilizers and biofortifiers.” Patent BR1020230164439.

## Figures and Tables

**Figure 1 plants-14-01495-f001:**
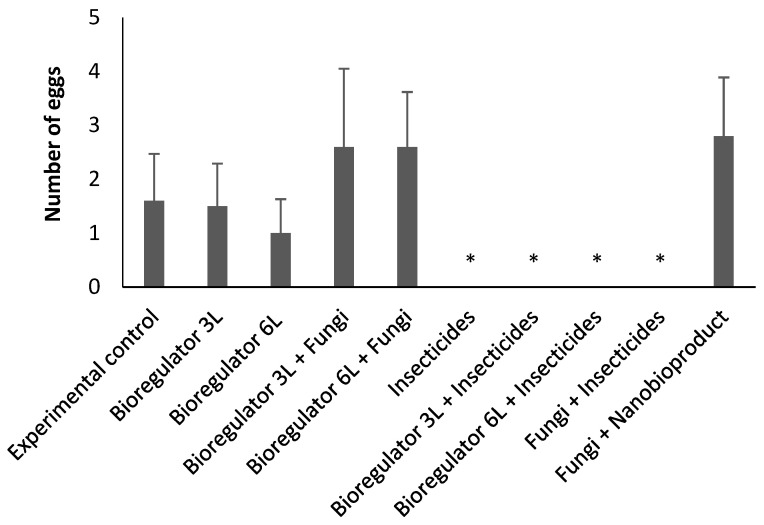
Mean numbers of eggs of *Hypothenemus hampei* after contact with different treatments, consisting of combinations of chemical ethylene synthesis inhibitor, chemical and fungi-based insecticides, and ethylene synthesis inhibitor nanobioproduct after 30 days. The asterisk (*) denotes variance, mean, and standard deviation values of the sample assumed to be zero.

**Figure 2 plants-14-01495-f002:**
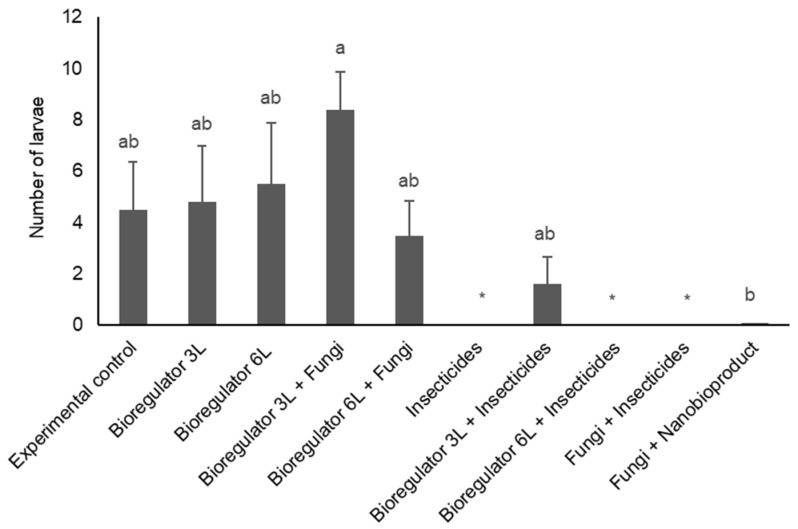
Mean numbers of larvae of *Hypothenemus hampei* after contact with different treatments, consisting of combinations of chemical ethylene synthesis inhibitor, chemical and fungi-based insecticides, and ethylene synthesis inhibitor nanobioproduct after 30 days. Bars topped with the same letters indicate no significant difference (*p* > 0.05). The asterisk (*) indicates treatments that differ statistically from the control.

**Figure 3 plants-14-01495-f003:**
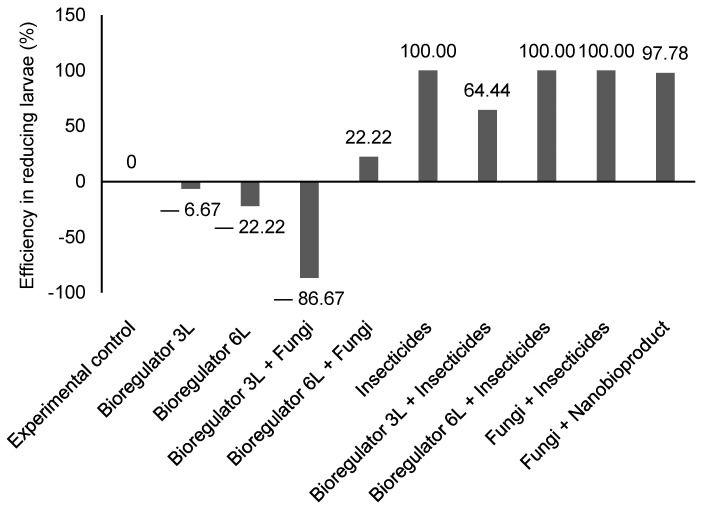
Efficiency of reduction (%) of *Hypothenemus hampei* larvae in response to different treatments, consisting of combinations of chemical ethylene synthesis inhibitor, chemical and fungi-based insecticides, and ethylene synthesis inhibitor nanobioproduct after 30 days.

**Figure 4 plants-14-01495-f004:**
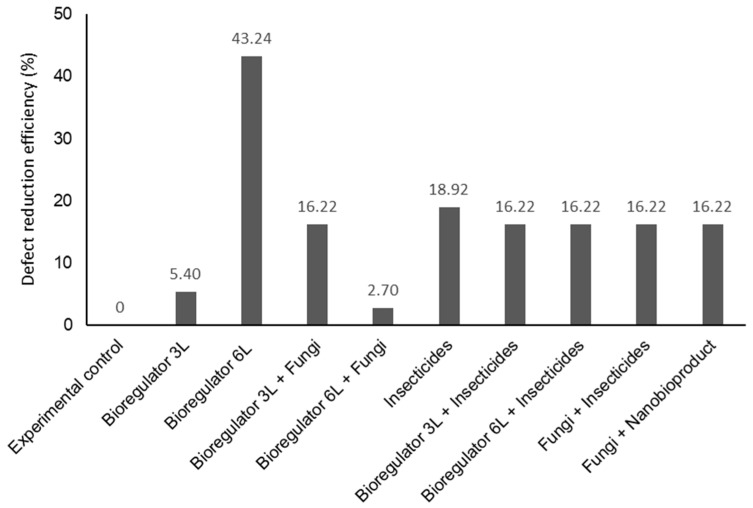
Reduction efficiency (%) of coffee bean defects from plants treated in the field with combinations of chemical ethylene synthesis inhibitor, chemical and fungi-based insecticides, and ethylene synthesis inhibitor nanobioproduct.

**Figure 5 plants-14-01495-f005:**
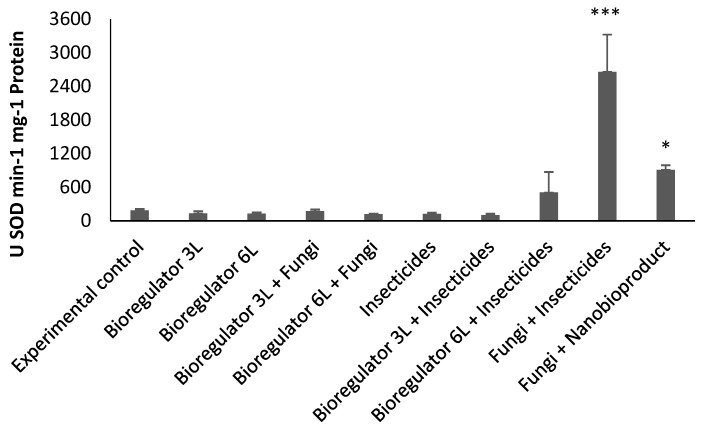
Enzymatic activity of superoxide dismutase (SOD) in coffee beans from plants treated in the field with combinations of chemical ethylene synthesis inhibitor, chemical and fungi-based insecticides, and ethylene synthesis inhibitor nanobioproduct. Asterisks indicate statistically significant differences compared to the control: *p* < 0.05 (*) and *p* < 0.001 (***).

**Figure 6 plants-14-01495-f006:**
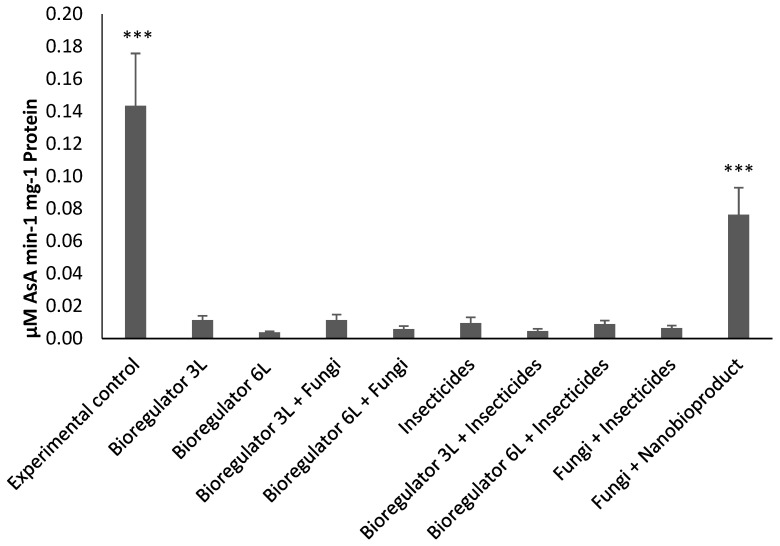
Enzymatic activity of ascorbate peroxidase (APX) in coffee beans from plants treated in the field with combinations of chemical ethylene synthesis inhibitor, chemical and fungi-based insecticides, and ethylene synthesis inhibitor nanobioproduct. Asterisks indicate statistically significant differences compared to the control: *p* < 0.001 (***).

**Figure 7 plants-14-01495-f007:**
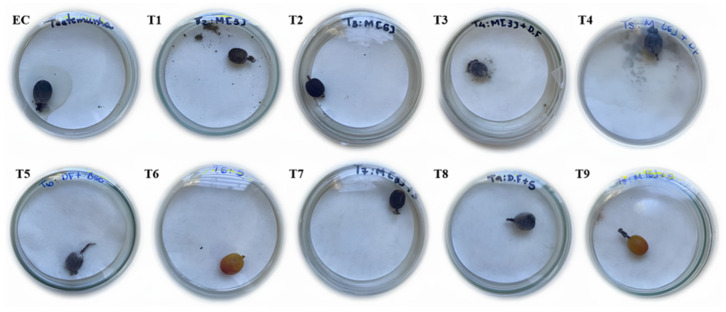
In vitro bioassay evaluating the combination of different products for controlling *Hypothenemus hampei*: (EC)—experimental control; (T1)—bioregulator 3 L ha^−1^; (T2)—bioregulator 6 L ha^−1^; (T3)—bioregulator 3 L ha^−1^ + fungi; (T4)—bioregulator 6 L ha^−1^ + fungi; (T5)—insecticides; (T6)—bioregulator 3 L ha^−1^ + insecticides; (T7)—bioregulator 6 L ha^−1^ + insecticide; (T8)—insecticides + fungi; and (T9)—fungi + nanobioproduct.

**Table 1 plants-14-01495-t001:** Physical and sensory analysis of arabica coffee beans subjected in the field to different treatments consisting of combinations of chemical ethylene synthesis inhibitor, chemical and fungi-based insecticides, and ethylene synthesis inhibitor nanobioproduct.

Treatment	Note	Acid	Body	Sweetness	Finish	Defect	Type
EC	83.50	5.89	6.22	6.11	5.78	104.83 a	5.57
T1	85.33	6.89	7.11	7.44	7.11	99.17 ab	5.67
T2	84.56	6.44	6.67	6.56	6.33	59.50 b	4.67
T3	83.11	5.89	5.89	5.89	5.44	87.83 ab	5.67
T4	81.50	5.33	5.78	5.44	4.56	102.00 a	6.00
T5	85.22	6.78	7.22	7.22	6.67	85.00 ab	5.67
T6	83.33	6.11	6.33	6.22	5.67	87.83 ab	6.00
T7	83.50	6.00	6.22	6.22	5.56	70.83 ab	5.00
T8	83.11	5.89	5.89	5.44	5.44	76.50 ab	5.33
T9	83.11	5.89	6.00	5.78	5.33	99.17 ab	5.67

Means followed by different letters differ by Tukey’s test. (EC)—experimental control; (T1)—bioregulator 3 L ha^−1^; (T2)—bioregulator 6 L ha^−1^; (T3)—bioregulator 3 L ha^−1^ + fungi; (T4)—bioregulator 6 L ha^−1^ + fungi; (T5)—insecticides; (T6)—bioregulator 3 L ha^−1^ + insecticides; (T7)—bioregulator 6 L ha^−1^ + insecticide; (T8)—insecticides + fungi; (T9)—fungi + nanobioproduct.

**Table 2 plants-14-01495-t002:** Treatments composed of a chemical ethylene synthesis inhibitor, insecticides, entomopathogenic fungi, and ethylene synthesis inhibitor nanobioproduct used in CBB management under in vitro and field conditions.

Treatments	Composition
EC	Experimental control
T1	Bioregulator 3 L ha^−1^
T2	Bioregulator 6 L ha^−1^
T3	Bioregulator 3 L ha^−1^ + Fungi
T4	Bioregulator 6 L ha^−1^ + Fungi
T5	Insecticides
T6	Bioregulator 3 L ha^−1^ + Insecticides
T7	Bioregulator 6 L ha^−1^ + Insecticides
T8	Insecticides + Fungi
T9	Fungi + Nanobioproduct

(EC)—Distilled water; (T1) Mathury^®^ (3 L ha^−1^); (T2) Mathury^®^ (6 L ha^−1^); (T3) Mathury^®^ (3 L ha^−1^) + DuoFunghi^®^; (T4) Mathury^®^ (6 L ha^−1^) + DuoFunghi^®^; (T5) Sperto^®^; (T6) Mathury^®^ (3 L ha^−1^) + Sperto; (T7) Mathury^®^ (6 L ha^−1^) + Sperto; (T8) Sperto^®^ + DuoFunghi^®^; (T9) Duofunghi^®^ + *B. subtilis* coated with zinc oxide nanoparticles.

## Data Availability

The original contributions presented in the study are included in the article. Further inquiries can be directed to the corresponding authors.

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
