# Peer review of "Combination of Nanobioproduct and Chemical Ethylene Synthesis Inhibitor with Entomopathogenic Fungi: A Novel Management Strategy for Coffee Berry Borer in Arabica Coffee"

_plants, 2025, doi:10.3390/plants14101495_

Round 1

Reviewer 1 Report

Comments and Suggestions for Authors

I have given my comments in the attached file.

Author Response

Dear Reviewer,
Below is the file with the answers attached.

Reviewer 2 Report

Comments and Suggestions for Authors

The authors conducted a study on the combination of a nanobioproduct and a chemical ethylene-synthesis inhibitor with entomopathogenic fungi as a novel management strategy for coffee berry borer in Arabica coffee. The research has significant scientific value; however, there are several critical issues that require clarification and verification.

Figure 1: What statistical method did the authors use for significance analysis in Figure 1? The figure suggests no significant differences, which seems counterintuitive. Some treatments resulted in almost no oviposition by CBB, yet there is reportedly no significant difference between these groups and others with high egg numbers. This requires clarification.

Additionally, is the "Experimental control" group untreated (without insecticide)? If so, how could there still be no significant difference? The absence of significance here also seems questionable.

Figure 2:Since Figure 1 clearly shows no significant differences in the number of eggs across treatments, the calculated “egg reduction efficiency” in Figure 2 is statistically invalid or misleading. This calculation ignores variance and statistical significance. Reduction efficiency should only be reported when significant differences have been demonstrated in the egg counts.

Figure 3: the Experimental control group is marked as "ab," while the insecticide treatments (T5, T7, T8, and T9) are marked as "b." This indicates that there is no significant difference between the control and these treatments. Therefore, the conclusion that these treatments “significantly reduced” the number of larvae is not supported by the statistical results presented.

Lines 152–156 In Table 1, only treatment T2 (b) seems to be significantly different from the Experimental control (a) in terms of defect counts. Other groups do not show clear significant differences. Therefore, stating that “treatments affected defects” is inaccurate. The conclusion should be that only certain treatments (e.g., T2) showed a significant reduction in defects compared to the control.

Regarding the "Finish" parameter in Table 1, if this refers to "aftertaste," it should be explicitly explained. Not all readers may be familiar with coffee sensory evaluation terminology.

Figure 5: Error bars should be included in Figure 5, and proper significance analysis should be performed. The term “superior” can only be used if statistical significance is demonstrated.

Figure 6: Considering the error bars shown in Figure 6, is there really no significant difference? This again seems counterintuitive and requires clarification.

Lines 211–215: The conclusion drawn here lacks statistical support. Without proper significance testing, such conclusions are unjustified, especially when no significant differences are detected.

The discussion should be based on actual statistically significant differences. If no significance exists, many of the claims made in the discussion are unfounded.

Lines 448–466: For comparing the efficiencies between multiple treatment groups (as in Figure 5), Abbott's formula alone is insufficient. Statistical analysis (such as ANOVA or non-parametric tests) based on the raw data is required to validate these efficiency comparisons.

Furthermore, the LSD test assumes normal distribution and homogeneity of variances. It is not suitable for data analyzed with GLM models. The authors need to carefully check whether the statistical tests applied are appropriate.

Author Response

(The authors gave the same response as above.)

Round 2

Reviewer 1 Report

Comments and Suggestions for Authors

1) The authors don't want to disclose the detailed protocol of green synthesis of NPs due to an ongoing patent application. However, providing each protocol in detail is very essential for the readers interest and for reproducibility. If the authors has obtained provisional patent number, then it seems ok to disclose each and every information in detail. 

2) Why did you choose HCl for bacterial lysis, as HCl caused NPs digestion and dissolution, which can provide wrong interpretation of results?

3) Based on spectrum, how did you estimate an average nanoparticle size of 2–4 nm? What are the calculations behind it? Please provide detailed method. I will suggest you to perform the TEM of the extracted NPs to know the NPs size. 

4) Authors mentioned that it's difficult to perform the zeta potential of NPs trapped on bacteria. But now they are able to extract the NPs via bacterial lysis. So, now it seems feasible to perform the zeta potential of these extracted NPs.

Author Response

Lavras, May 09th, 2025

Dear Editor,

On behalf of all fellow authors, I am pleased to submit a new version of our manuscript entitled “ Combination of nanobioproduct and chemical ethylene-synthesis inhibitor with entomopathogenic fungi: a novel management strategy for coffee berry borer in arabica coffee” by Sousa et al., for your consideration for publication in Plants, Special issue: Biosynthesis of Nanoparticles: A New Approach Between Nanotechnology and Plant Bioinputs.

We are grateful for the time and effort the reviewers spent on our manuscript, and we have carefully considered their suggestions to improve the quality of the paper. We have incorporated the accepted points in the revised manuscript, highlighting all changes in yellow for your convenience. All the raised issues are clarified below.

Please do not hesitate to let us know if you require any additional assistance or clarification. We thank you for your kind attention. 

Sincerely yours, 

Joyce Dória, Ph.D. 

Professor 

Federal University of Lavras (UFLA) 

Lavras, Brazil 

E-mail: joyce.doria@ufla.br

Reviewer 2 Report

Comments and Suggestions for Authors

The authors have answered my questions, I have no other comments.

Author Response

Dear reviewer,

We acknowledge the feedback provided. We are submitting a new review version with some further clarifications requested by reviewer 1 (Sousa et al. (2025)_track changes3).

Round 3

Reviewer 1 Report

Comments and Suggestions for Authors

The authors has answer all the raised comments.